

# Sensitivity analysis of the meteorological pre-processor MPP-FMI 3.0 using algorithmic differentiation

John Backman[1], Curtis Wood[1], Mikko Auvinen[1,2], Leena Kangas[1], Hanna Hannuniemi[1], Ari

Karppinen[1], and Jaakko Kukkonen[1]

[1]Atmospheric Composition Research, Finnish Meteorological Institute, Helsinki, Finland

[2]Department of Physics, Division of Atmospheric Sciences, University of Helsinki, Helsinki,

Finland

*Correspondence to*: John Backman ([john.backman@fmi.fi](mailto:john.backman@fmi.fi))

**Abstract.** The meteorological input parameters for urban and local scale dispersion models can be
evaluated by pre-processing meteorological observations, using a boundary-layer parametrization

model. This study presents a sensitivity analysis of a meteorological pre-processor model (MPP-
FMI) that utilises readily available meteorological data as input. The sensitivity of the pre-processor
to meteorological input was analysed using algorithmic differentiation (AD). The AD tool used was
TAPENADE. The AD method numerically evaluates the partial derivatives of functions that are
implemented in a computer program. In this study, we focus on the evaluation of vertical fluxes in

the atmosphere, and in particular on the sensitivity of the predicted inverse Obukhov length and
friction velocity on the model input parameters. The study shows that the estimated inverse
Obukhov length and friction velocity are most sensitive to wind speed, and second most sensitive to
solar irradiation. The dependency on wind speed is most pronounced at low wind speeds. The
presented results have implications for improving the meteorological pre-processing models. AD is

shown to be an efficient tool for studying the ranges of sensitivities of the predicted parameters on
the model input values quantitatively. A wider use of such advanced sensitivity analysis methods
could potentially be very useful in analysing and improving the models used in atmospheric
sciences.

## 1 INTRODUCTION

Any urban or local scale dispersion model requires specific information about the state of the
atmospheric boundary layer (ABL) as input values. This information can be estimated from
available meteorological observations by so-called meteorological pre-processors (e.g., Van Ulden



and Holtslag, 1985). This allows for the use of advanced meteorological input data into the models, even when no atmospheric turbulence measurements would be available. These evaluations are commonly done by applying an energy-flux method that estimates turbulent heat and momentum fluxes in the boundary layer to derive desired boundary-layer scaling parameters (e.g., Fisher et al., 2001; Van Ulden and Holtslag, 1985).

The urban scale dispersion models at the Finnish Meteorological Institute (FMI) rely on advanced meteorological input from a meteorological pre-processor that is mainly based on the boundary-layer parametrization of Van Ulden and Holtslag (1985). These dispersion models include, e.g., a Gaussian road network dispersion model (CAR-FMI, Kukkonen et al., 2001; Kauhaniemi et al., 2008) and an urban multiple source Gaussian dispersion model (UDM-FMI, Karppinen et al., 2000b). The models are used to model emissions, dispersion and transformation of pollution for urban areas.

Model sensitivity studies can be done using algorithmic differentiation (AD), which is a technique to compute partial derivatives by differentiation of the functions and operations that comprise computer programmes. In this study a source transformation AD tool called TAPENADE (Hascoët and Pascual, 2013) is employed to differentiate the procedures of a meteorological pre-processor. TAPENADE was chosen because it is the only Fortran source transformation tool that is free for academic use, actively supported and developed, and is well documented.

In essence, an AD tool will produce a differentiated set of the equations of a code, based on the sequence of operations that the computer program comprise. The differentiated code will also compute, in addition to the original outputs, the partial derivatives of the outputs with respect to the pre-processors inputs at machine precision. In the source transformation method of AD, an additional set of statements is added (in text) to the computer program that propagates the derivative information through the computer program. In this way, a standard (Fortran in this case) compiler can be used which is not the case for the other AD methods (such as operator overloading and AD enabled compilers).

AD has applications that span multiple disciplines of science such as engineering, physics, chemistry, medicine, where it can be used for e.g. sensitivity analyses, optimisation, and inverse problem solving, etc. (Griewank and Walther, 2008). In fact, AD has applications wherever partial derivatives of computer programmes can be made useful. It is not the intention to give a full literature review of research that has benefited from AD but rather a brief overview of its applications in geophysical research and in particular using TAPENADE.





The AD tool TAPENADE has been used for a variety of different physics models as follows. A general purpose atmospheric radiative transfer model for remote sensing applications made use of the superior numerical accuracy of AD, in comparison to finite difference perturbations, for evaluation of satellite trace gas spectra (Schreier et al., 2014). Moreover, the AD method was later recommended for the same model due to lower computational cost and greater numerical accuracy

when solving non-linear inverse radiative transfer problem through iteration (Schreier et al. 2015). A meteorology–chemistry coupled model also made use of AD source transformation when developing a four-dimensional variational data assimilation procedure for the model (Guerrette and Henze, 2015). TAPENADE has also been used for a sensitivity study of a sea-ice model to determine optimal model parameters in a minimisation algorithm (Kim et al., 2006). More

information and literature on AD can be found through the community driven portal for algorithmic differentiation (www.autodiff.org).

The sensitivity on input data of the above mentioned meteorological pre-processing method has not previously been systematically investigated. The aim of this study is to quantitatively determine the sensitivities of meteorological output parameters on model input for the meteorological pre-

processor MPP-FMI (Karppinen et al., 1997, 2000a). This procedure is useful for analysing in detail the functioning of the computer program corresponding to the model MPP-FMI. The modelled sensitivities can also be compared to what would be physically feasible, based on a consideration of the relevant atmospheric processes. This will provide a useful additional test regarding the correct functioning of the computer code and the numerical procedures of the MPP-FMI model. Such a

thorough and quantitative sensitivity analysis also provides new information and insights regarding the further refinement of such models.

## 2  METHODS

### 2.1  The meteorological pre-processor MPP-FMI

The meteorological pre-processor is used to estimate turbulent fluxes, atmospheric stability, and

boundary-layer scaling parameters based on meteorological observations at fixed locations. The scope of this study is to determine the sensitivity of this model for deriving the vertical fluxes in the boundary layer. However, we have not addressed the the routines within the MPP-FMI model that deal with radiosonde data, to estimate the convective velocity scale (i.e. Deardorff velocity), vertical temperature gradient, and mixing height. The scope of the study is depicted in Fig. (1).

The meteorological observations used by the MPP-FMI model as input comprise temperature ($T_2$),



wind speed ($U$) and wind direction at a height of 10 m, amount of predominant clouds ($C_C$), cloud height ($C_Z$), sunshine fraction, state of the ground (wet, dry, snow, ice etc.), and precipitation. These are needed by the pre-processor in order to model boundary-layer scaling parameters required by the urban scale dispersion models.

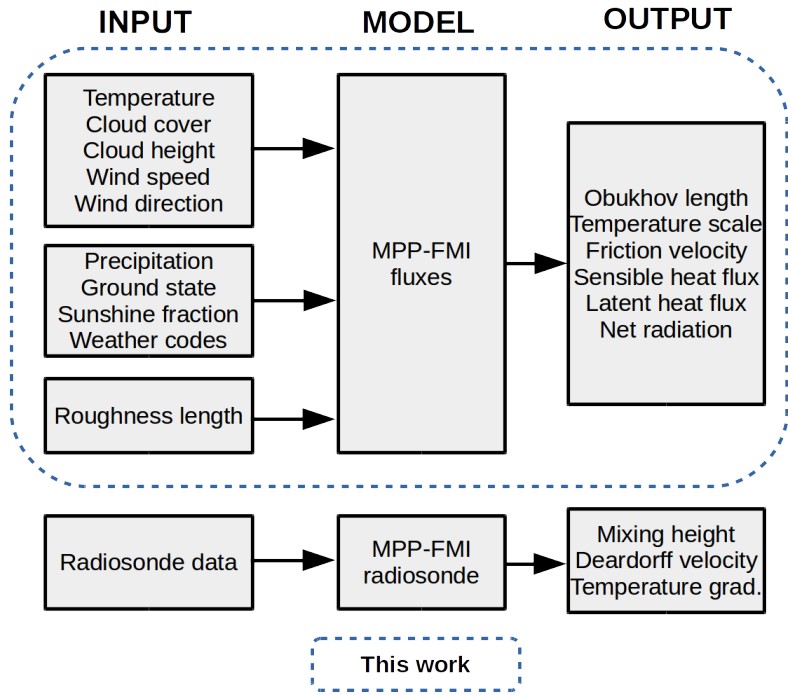

**Figure 1:** A schematic diagram on the flow of information of the meteorological pre-processor MPP-FMI.

MPP-FMI is originally based on the work by Van Ulden and Holtslag (1985) with modifications that makes the parametrisation more suitable for high latitudes and urban areas (Karppinen et al., 1997, 2000a). Central to this method is the surface heat-budget equation

$$Q* - Q_G = Q_H + Q_E \quad . \qquad (1)$$

In Eq. (1), $Q*$ is the surface net radiation, $Q_G$ is the soil heat flux, $Q_H$ is the sensible heat flux and $Q_E$ is the latent heat flux. The terms that comprise Eq. (1) are not commonly available from
measurements (although there are measurements of eddy-covarince at some research sites; Wood et al., 2013) and are therefore estimated by the meteorological pre-processor. A comprehensive description of MPP-FMI is already available in literature (Karppinen et al., 1997). However, a brief





overview of the model structure will be presented in the following for convenience.

First, the meteorological pre-processor estimates available energy $Q^*–Q_G$ by decomposing the terms

into components of (i) net shortwave radiation using incoming shortwave radiation and albedo, (ii) net longwave radiation from surface radiative temperature and cloud-base radiation temperature (specific for MPP-FMI) using a constant dry adiabatic lapse-rate and cloud-base height, and (iii) estimated heat flux into the ground from estimated temperature difference between the ground and a reference height of 50 metres. Then, the term $Q_E$ is estimated using a simplified Penman-Monteith

equation (Van Ulden and Holtslag, 1985). Consequently, an estimate of the sign of $Q_H$ is obtained which will determine if the subsequent calculations are to be done using stability functions for stable or unstable conditions.

According to surface-layer similarity theory, both friction velocity ($u_*$) and temperature scale for turbulent heat transfer ($\theta_*$) can be expressed as vertical profiles. For $u_*$, which is a measure of the

surface production of turbulent kinetic energy, the equation is

$$u_* = \frac{U(z)\,k}{\ln\left(\dfrac{z}{z_0}\right) - \psi_M\left(\dfrac{z}{L}\right) + \psi_M\left(\dfrac{z_0}{L}\right)} \quad . \tag{2}$$

In Eq. (2), $U$ is wind speed at height $z$, $z_0$ is the surface roughness length, $k$ is the von Karman constant, and the terms $\psi_M$ are stability functions; see Appendix A for details. $L$ is the Obukhov length which is an atmospheric stability measure that describes the relative importance of surface production of turbulence due to shear stress and buoyancy forces.

Similarly to $u_*$, $\theta_*$ can be written as

$$\theta_* = \frac{k\left[\theta(z_2) - \theta(z_1)\right]}{\ln\left(\dfrac{z_2}{z_1}\right) - \psi_H\left(\dfrac{z_2}{L}\right) + \psi_H\left(\dfrac{z_1}{L}\right)} \quad , \tag{3}$$

where $z_1$ and $z_2$ are arbitrary heights in the surface layer, $\theta$ is the potential temperature at the respective heights, and the terms $\psi_H$ are stability functions. Both Eqs (2 and 3) and their respective stability functions are used as described in Van Ulden and Holtslag (1985). Using Eq. (3), $\theta(z_2)$ at a reference height of 50 m can be modelled from measurements of $\theta(z_1)$. This is done by solving $\theta_*$

from the definition of $L$

$$L = \frac{u_*^2\,\theta}{k\,g\,\theta_*} \tag{4}$$

and substituting it into Eq. (3). In Eq. (4) $g$ is the acceleration due to gravity. This completes the



modelling of $\theta_*$ using surface-layer similarity theory using the profile method (Van Ulden and Holtslag, 1985).

In addition to Eqs (3) and (4), $\theta_*$ can also be estimated using the energy-budget method derived
from the modified Penman-Monteith equation

$$\theta_* = \left( \frac{\alpha S}{S+1} - 1 \right)\left( \frac{Q^* - G}{\rho\, c_p u_*} \right) + \alpha\, \theta_d \quad , \tag{5}$$

where $\alpha$ is the Priestley-Taylor moisture parameter, $S$ is the saturation enthalpy curve of water vapour, $\rho$ the density of air, $c_p$ is the specific heat capacity of air, and $\theta_d$ is an empirical temperature scale. The derivation of Eq. (5) is done using the equations in Van Ulden and Holtslag (1985). In MPP-FMI, however, the parametrisation of $S$ is different from that of Van Ulden and Holtslag
(1985) in order to extend the temperature range of the parametrisation. Both parametrisations are very similar and are solely functions of surface temperature.

Finally, the value for $L$ is found iteratively by changing $L$ until $\theta_*$ from the profile method is equal to $\theta_*$ from the energy-budget method of Eq. (5); namely Eq. (5) is equal to $u_*^2\, \theta/(k\, g\, L)$. This iteration will consequently impact $u_*$ and $\theta_*$ as described above. In addition, $Q^*$, $G$, $Q_H$, and $Q_E$ will
also change during the iteration because of the stability functions of Eqs (2) and (3).

## 2.2 Algorithmic differentiation

Algorithmic differentiation (AD) deals with the numerical evaluation of derivatives of functions that are implemented in a computer programme. Any computer program, no matter how complex, performs a sequence arithmetic operations (addition, subtraction, division, etc.) or elementary
functions (exponential, trigonometric, etc.) whose derivatives are known. AD exploits this fact by applying the chain rule of differentiation to the entire sequence of operations within the program (Griewank and Walther, 2008). This systematic approach yields numerical derivative values at machine precision, which describe how the program's results (i.e. outputs) depend on its inputs. The AD method performs each differentiation operation at machine precision and does not employ
approximate techniques, such as finite differences. For this reason AD does not suffer from truncation or round-off errors.

AD is further separated into two modes, a forward mode or a reverse mode (Griewank and Walther, 2008). Here the discussion will be limited to the forward mode, which has been employed in this study. As a starting point, consider an arbitrary computer program that takes $n$ input variables and
returns $m$ outputs. It can be described as a vector-valued function



$$y = F(\boldsymbol{x}) \quad , \tag{6}$$

such that, the function $F$ maps $\mathbb{R}^n \rightarrow \mathbb{R}^m$ where $\boldsymbol{x} \in \mathbb{R}^n$ defines the input and $\boldsymbol{y} \in \mathbb{R}^m$ the output vectors.

Application of the forward mode AD to Eq. (6) yields a new implementation of the program, which, in addition to the original function evaluation, evaluates its differential

$$\dot{\boldsymbol{y}}_k = F'(\boldsymbol{x})\dot{\boldsymbol{x}}_k \quad . \tag{7}$$

In Eq. (7), $F'(\mathbf{x}) \in \mathbb{R}^{m \times n}$ defines the Jacobian matrix, which contains all first-order partial derivatives $\partial \boldsymbol{y} / \partial \boldsymbol{x}$ and $\dot{\boldsymbol{x}}_k = (\partial x_1 / \partial x_k, \dots, \partial x_k / \partial x_k, \dots, \partial x_n / \partial x_k)^T$ is the seeding vector, which can be viewed as the $k^{\text{th}}$ unit vector that operates on the Jacobian. The result is the $k^{\text{th}}$ column from the Jacobian matrix $\dot{\boldsymbol{y}}_k = (\partial y_1 / \partial x_k, \partial y_2 / \partial x_k, \dots, \partial y_m / \partial x_k)^T$ which yields the dependency of all outputs with respect to the user-specified $x_k$ input parameter. In the forward mode differentiated

computer program, the derivative evaluations based on the chain rule contained in Eq. (7) are performed following the same order as the associated operations in Eq. (6), but always such that the derivative operations are executed after their corresponding step in the original program have completed.

A typical goal in sensitivity analysis is to obtain the full Jacobian. Utilizing forward mode AD, this

is achieved by repeating the computation of Eq. (7) *n* times to yield all the columns of the Jacobian matrix. This is best illustrated with an example matrix (Eq. 8) where the first column of the Jacobian is chosen. Thus, for a given input **x** one can construct the Jacobian using AD and extract the derivatives of the output of interest at that point. This procedure can then be repeated for any number of points.

$$\dot{\boldsymbol{y}}_1 = \underbrace{\begin{bmatrix} \dfrac{\partial y_1}{\partial x_1} \\ \dfrac{\partial y_2}{\partial x_1} \\ \vdots \\ \dfrac{\partial y_m}{\partial x_1} \end{bmatrix}}_{\boldsymbol{y}_{k=1} \in \mathbb{R}^m} = \underbrace{\begin{bmatrix} \dfrac{\partial y_1}{\partial x_1} & \dfrac{\partial y_1}{\partial x_2} & \cdots & \dfrac{\partial y_1}{\partial x_n} \\ \dfrac{\partial y_2}{\partial x_1} & \dfrac{\partial y_2}{\partial x_2} & \cdots & \dfrac{\partial y_2}{\partial x_n} \\ \vdots & \vdots & \ddots & \vdots \\ \dfrac{\partial y_m}{\partial x_1} & \cdots & \cdots & \dfrac{\partial y_m}{\partial x_n} \end{bmatrix}}_{F'(\boldsymbol{x}) \in \mathbb{R}^{m \times n}} \underbrace{\begin{bmatrix} 1 \\ 0 \\ \vdots \\ 0 \end{bmatrix}}_{\dot{\boldsymbol{x}}_{k=1} \in \mathbb{R}^n} \tag{8}$$

The reverse mode of AD is not applied in this work because the number of input variables are roughly the same as the number of output variables ($m \approx n$). The reverse mode should be favoured when $n \gg m$ (Griewank and Walther, 2008). Again, the differentiation was performed using the AD





tool called TAPENADE (Hascoet and Pascual, 2013). TAPENADE has been developed by the French National Institute for computer science and applied mathematics (Inria) and is free-of-
charge through a web-based user interface.

## 3   RESULTS

Input parameters that are used in table lookups are in this work replaced by the parameters that are the outcome of the table lookup (Appendix B). Namely, precipitation and state-of-the-ground input data are used in a table lookup to estimate a value for the Priestley-Taylor moisture parameter $\alpha$,
whereas state-of-the-ground is used to estimate the surface albedo ($r$). From a sensitivity study point-of-view, it makes more sense to be able to assess the sensitivity to $\alpha$ and $r$ directly, rather than the sensitivity of the table lookup procedure. Therefore, in this work, the table lookup variables $r$ and $\alpha$ are included as inputs to the MPP-FMI, which also reduces the number of input variables to be analysed. Thus, the sensitivity analysis becomes more straightforward to interpret because
inherent step-functions of table lookups are circumvented.

In addition to replacing the table lookup with parameters that result from the lookups, the sunshine fraction has been replaced with net incoming solar radiation at the surface ($R_S$). Replacing the sunshine fraction with $R_S$ is motivated by an increased availability of direct measurements of $R_S$. Originally the sunshine fraction is used in a regression to derive $R_S$ (Karppinen et al., 1997).

### 3.1  Obukhov length sensitivity

We have selected the ranges of the input parameters for the sensitivity analysis to be the commonly occurring ones in the meteorological and environmental conditions in the city of Helsinki, Finland. For instance, the ambient temperatures were assumed to range from -20 °C to + 30 °C. These ranges have been presented in Table 1.

**Table 1.** Range of parameters used for studying the sensitivity of $L^{-1}$. For each range, six points were linearly spaced within the range. This amounts to $6^8$ (1.7 million) combinations of input variables to be evaluated; resulting in $6^8$ Jacobian matrices. In the table, $z_0$ is the roughness length, $r$ is the surface albedo, $T_2$ is the temperature at the height of two metres, $C_C$ is the cloud cover, $U$ is the wind speed at 10 m, $\alpha$ is the Priestley-Taylor moisture parameter and $R_S$ is the solar irradiance.

| Inputs | $z_0$ [m] | $r$ | $T_2$ [ºC] | $C_C$ | $C_Z$ [m] | $U$ [m s$^{-1}$] | $\alpha$ | $R_S$ [W m$^{-2}$] |
|---|---|---|---|---|---|---|---|---|
| Range | 0.3–1.3 | 0.05–0.7 | -20–30 | 0–1 | 30–6000 | 1–20 | 0.5–1.0 | 0–900 |






The values in Table 1 were then used to construct the Jacobian (Eq. 8) for every combination of the meteorological input variables. The rows of interest for this work are those rows in the Jacobian containing the sensitivity information of $L^{-1}$ and $u_*$ since these are further needed in the Gaussian dispersion models CAR-FMI and UDM-FMI. In addition to $L^{-1}$ and $u_*$, the Jacobian comprise

sensitivity information for the quantities $Q_H$, $Q_E$, $Q^*$, and $\theta_*$ to the respective input variables listed in Table 1.

The range and units of the input variables varies greatly. Therefore, the inter-comparison of partial derivatives of the outputs with respect to the input data as such is not desirable. In order to make the partial derivatives inter-comparable, the partial derivatives have been normalized by 10% of the

input range of the respective input variables denoted $\Delta x_i$. The range of the input data is listed in Table 1.

In Fig. (2), the sensitivity of the inverse Obukhov length ($L^{-1}$) is shown for all combinations of the input parameters listed in Table 1. $L^{-1}$ describes the atmospheric stability. For neutral conditions $L^{-1}{\approx}0$. When $L^{-1}{<<}0$ the atmosphere is unstable, and when $L^{-1}{>>}0$ the atmosphere is stable. For

clarity, Fig. (2) is further separated into a low wind-speed situation with all other input variables varied (the main figure). The insert figure contains all combinations of input parameters associated with wind speeds in the range of 4–20 m s$^{-1}$. The figure is separated into a low and high wind speed situation because the model is much more sensitive to input data when the wind speed is low; $U{\approx}1$ m s$^{-1}$.

An obvious conclusion based on the results in Fig. (2) is that the wind speed $U$ is the most important parameter, and the solar irradiation $R_S$ is the second most important one, with respect to the predicted values of the inverse Obukhov length. This result could also be physically expected, since wind speed is the most obvious factor in terms of the formation of mechanical turbulence, whereas solar irradiation is a crucial parameter for the thermally induced turbulence.

As can be seen from Fig. (2), $L^{-1}$ is most sensitive to a change in $U$. When compared to the insert ($4{\leq}U{\leq}20$ m s$^{-1}$), the sensitivity to a change in wind speed is more pronounced at low wind speeds. When $L^{-1}$ is negative, which is the case of unstable and neutral conditions, the partial derivative $\partial L^{-1}/\partial U$ is positive. That means that an increase in $U$ will always favour the modelled stability to become more neutral. That is, a negative $L^{-1}$ and a positive partial derivative of $\partial L^{-1}/\partial U$ will tend to

move $L^{-1}$ towards neutral given that $U$ increases.





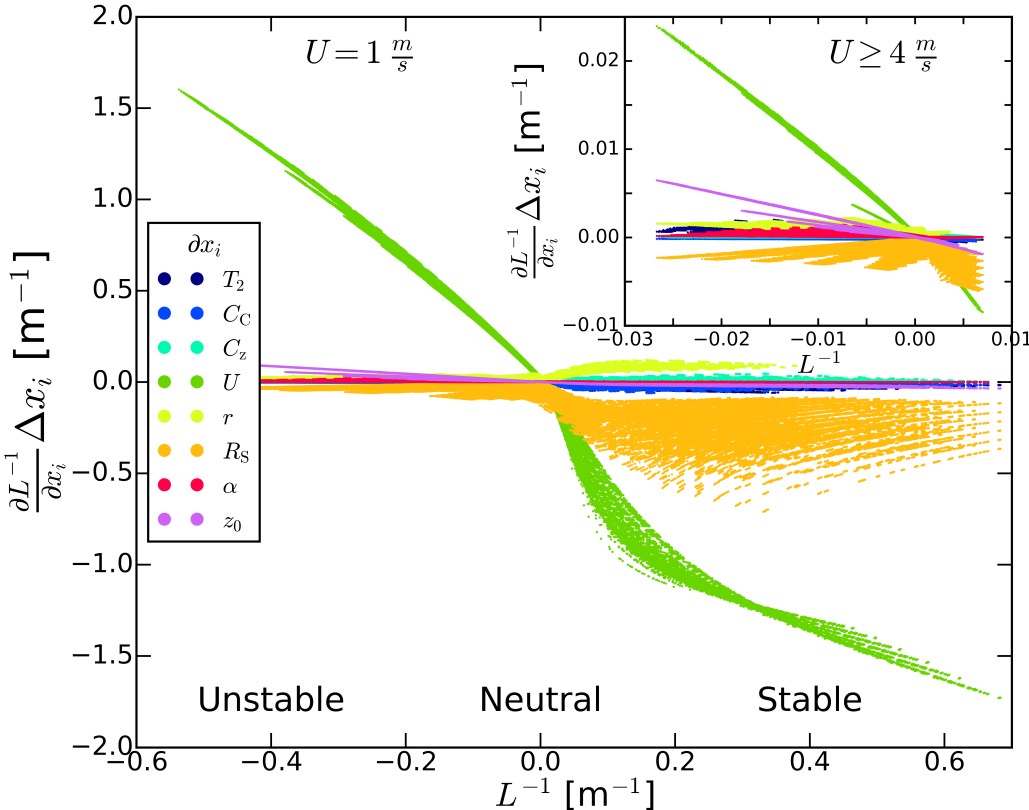

**Figure 2.** Sensitivity of inverse Obukhov length ($L^{-1}$) with respect to input variables of MPP-FMI. The main figure shows sensitivities to all the input variables when the wind speed ($U$) is 1 m s$^{-1}$. The insert shows sensitivities for wind speeds in the range of 4–20 m s$^{-1}$. In the figure, the partial derivatives have been normalised by the range of the input parameters ($\Delta x_i$) described in Table 1 in order to make them inter-comparable.

Conversely, when $L^{-1} > 0$ (i.e. stable to neutral), then $\partial L^{-1}/\partial U$ is always negative. This means that an increase in $U$ will therefore, again, tend to make $L^{-1}$ move towards neutral. This is in agreement with what one would expect in nature since an increase in $U$ will induce mechanical turbulence regardless of the initial stability and hence favour neutral conditions. At higher values of $U$, seen in

the insert of Fig (2), the $L^{-1}$ range in now restricted to roughly the range of -0.03–0.01 (i.e. neutral). The second most important input variable for the pre-processor with regard to $L^{-1}$ is $R_S$. The partial derivative $\partial L^{-1}/\partial R_S$ for all considered combination of input values remains exclusively negative, and even more so when $L^{-1} > 0$. This means that an increase in $R_S$ will always move the stability towards unstable. This follows the intuition that an increase in $R_S$ will increase buoyancy induced



turbulence, therefore favouring an unstable boundary layer. At low wind speeds, it has to be noted,
that the spread in the sensitivity of $L^{-1}$ to $R_S$, is an indication that other meteorological input
variables influence the results, especially when $L^{-1}>0$. This is evident from the fact that the
sensitivity to $R_S$ does not follow a single line, but is spread out. For example when $L^{-1}=0.3$ m$^{-1}$, then
$\partial L^{-1}/\partial R_S$ is in the range of -0.1–0.6 m$^{-1}$. The highest sensitivity to a change in $R_S$, at low wind
speeds, is when $R_S$ is close to zero and the surface albedo ($r$) is low. This information is, however,
not colour coded into the figure (so as not to degenerate the clarity of the figure).

### 3.2  Friction velocity sensitivity

The other important scaling parameter for the Gaussian models is $u_*$. Moreover, $u_*$ is also central for
the iteration procedure in the pre-processor when finding a value for $L^{-1}$. Table 2 summarizes the
input variable ranges for the $u_*$ sensitivity analysis. The variable range used for the sensitivity study
of $u_*$ differs from that of $L^{-1}$ in case of the selected wind speeds; the extremely high wind speeds
(from 12 to 20 m/s) have been omitted in case of the $u_*$ sensitivity analysis. The latter selection was
made in order to be able to present the results more clearly; the highest wind speeds also occur only
for a small fraction of time. The sensitivity of $u_*$ to different input variables is depicted in Fig. (3).
As for the corresponding results for $L^{-1}$, the wind speed $U$ was the most important parameter, and
the solar irradiation $R_S$ was the second most important one. This result is physically to be expected
also in case of the sensitivity of $u_*$.

**Table 2.** Range of parameters used for studying the sensitivity of $u_*$. Six points were linearly spaced within
the range, except for $U$ which comprise 10 logarithmically spaced points which amounts to roughly 2.8
million combinations of input variables. In the table, $z_0$ is the roughness length, $r$ is the surface albedo, $T_2$ is
the temperature at the height of two metres, $C_C$ is the cloud cover, $U$ is the wind speed at 10 metres, $\alpha$ is the
Priestley-Taylor moisture parameter and $R_S$ is the solar irradiance.

| Inputs | $z_0$ [m] | $r$ | $T_2$ [ºC] | $C_C$ | $C_z$ [m] | $U$ [m s$^{-1}$] | $\alpha$ | $R_S$ [Wm$^{-2}$] |
|---|---|---|---|---|---|---|---|---|
| Range | 0.3–1.3 | 0.05–0.7 | -20–30 | 0–1 | 30–6000 | 1–12 | 0.5–1.0 | 0–900 |

Amongst the input parameters, only $U$ and $z_0$ are present in the equation for $u_*$. The rest of the
sensitivity of $u_*$ is, to a varying degree, related to the cross sensitivity between $L^{-1}$ and $u_*$ through
Eqs (2-5). Since $u_*$ is a scaling parameter for the production of turbulent kinetic energy due to shear
stress, $u_*$ is generally high for high values of $U$. Thus, a generalisation can be made that $u_*$ is most
sensitive to $U$ at low wind speeds. Furthermore, the stability functions $\psi_M$ of Eq. (2) will increase $u_*$





the more negative (unstable) $L^{-1}$ becomes and decrease $u_*$ the more positive (stable) $L^{-1}$ becomes;

see Appendix A. For neutral stability ($L^{-1}{\approx}0$), the stability functions $\psi_M$ of Eq. (2) yield very similar

results for $u_*$. At higher wind speeds, the value of $z_0$ determines to a greater extent the sensitivity of

$\partial u_*/\partial U$. This is clearly visible when $u_*{>}1$ as six vertically separated groups of points in Fig. (3); six

groups because of six different values of $z_0$. This is, however, not colour coded into the figure so as

not to degenerate the clarity of the figure.

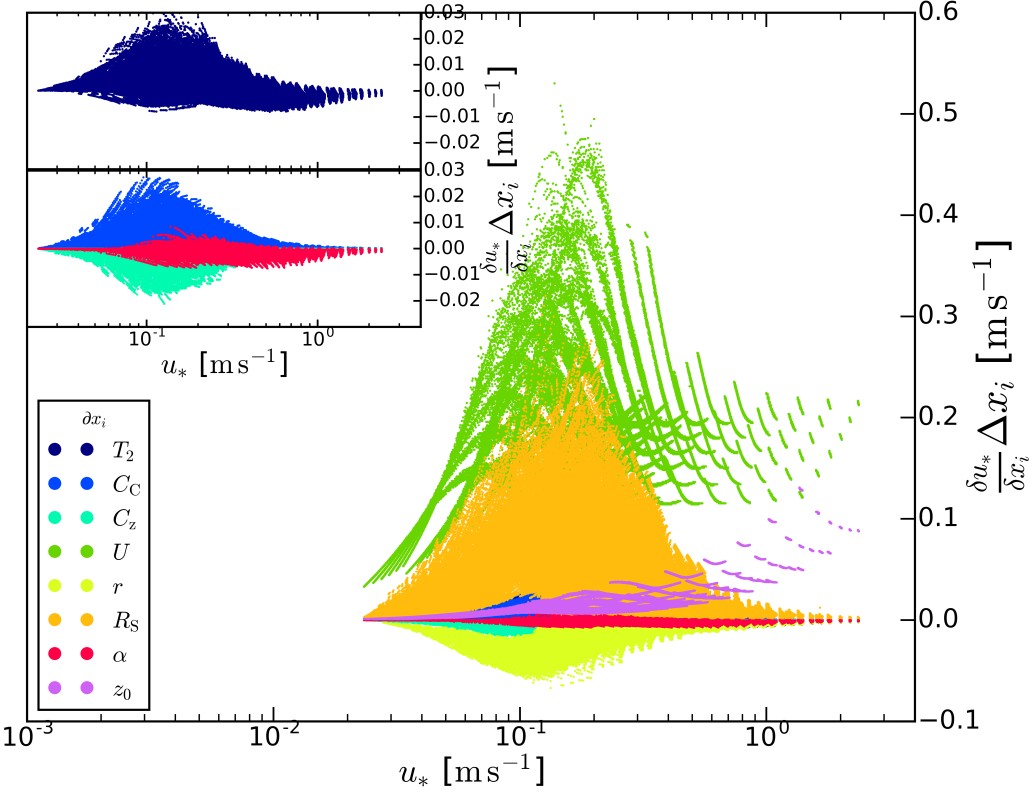

**Figure 3:** Sensitivity of friction velocity ($u_*$) with respect to input variables of MPP-FMI. The main figure
shows sensitivities of the most important input variables whereas the inserts show the less sensitive input
variables. The partial derivatives have been normalised by the range of the input parameters ($\Delta x_i$) described
in Table 2 in order to make them inter-comparable.

The second most important input parameter for $u_*$ is $R_S$. This holds true for low values of $u_*$. Based

on the discussion regarding the sensitivity of $L^{-1}$ this is expected. However, from Eq. (2) it is not

that clear that $u_*$ is sensitive to the solar radiation input into the pre-processor. Again, as $R_S$ changes,

this will impact the absolute values that comprise the energy budget equation; see Eq. (1). This in





turn will impact $\theta_*$ which consequently impacts $L^{-1}$ and ultimately $u_*$ through the stability functions.

However, at high $u_*$ the importance of $z_0$ will be more important for the modelled value of $u_*$ than $R_S$ as depicted in the figure. Opposite to the sensitivities to $U$, $R_S$ and $z_0$, an increase in surface albedo ($r$) will lower $u_*$ through $L^{-1}$.

### 3.3 Cross sensitivity

The sensitivity study of $L^{-1}$ and $u_*$ has shown that $U$ is the most important parameter for MPP-FMI.

$L^{-1}$ is highly sensitive to a change in $U$ when $U \approx 1\,\mathrm{m\,s^{-1}}$. Moreover, $u_*$ is also most sensitive to $U$. Because $u_*$ is a function of $L^{-1}$ (Eq. 2) and $L^{-1}$ is a function of $u_*$ (Eq. 4) these scaling parameters are interconnected. Thus, these scaling parameters are cross-sensitive.

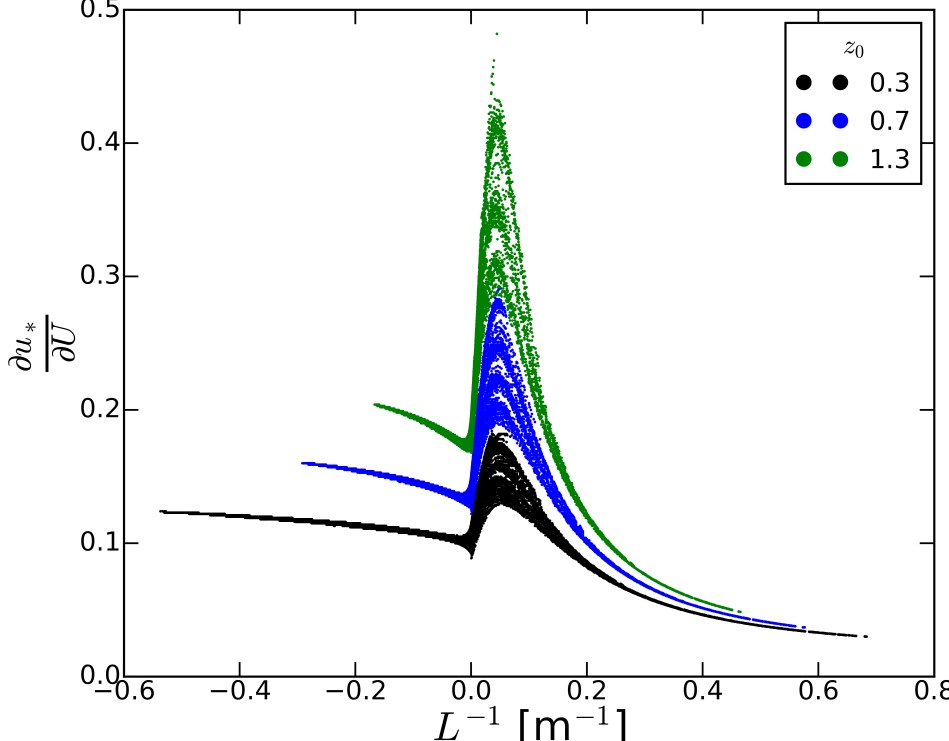

**Figure 4:** Cross sensitivity between atmospheric stability ($L^{-1}$) and friction velocity ($u_*$) with respect to wind speed ($U$) for different surface roughness lengths ($z_0$). Not all $z_0$ values in the range are plotted for clarity. Note that the y-axis of Fig. (4) is not scaled as in the previous figures because there is no inter-comparison between different input data in this figure.

Figure (4) shows the how the cross sensitivity between $\partial u_*/\partial U$ and $L^{-1}$. The figure shows that the





largest sensitivity of $\partial u_*/\partial U$ will be when $L^{-1}$ is around 0.1 m$^{-1}$; i.e. mildly stable. The different

behaviour of $\partial u_*/\partial U$ when $L^{-1}>0$ is likely due to the increased complexity of the stability functions ($\Psi_m$ in Eq. 2) for stable conditions than for unstable conditions; see Appendix A for details. This behaviour is not captured in Figs (2) and (3) although it could perhaps be inferred. This behaviour is also likely to be the case for real atmospheric conditions since a mildly stable boundary layer would be susceptible to increasing $U$ and consequently the production of wind shear induced turbulence

which would cause $u_*$ to increase. For highly stable conditions the sensitivity of $\partial u_*/\partial U$ levels out and is below the sensitivity for unstable conditions.

For unstable conditions ($L^{-1}\ll0$), the sensitivity of $\partial u_*/\partial U$ is less complex and the degree of sensitivity is largely dictated by $z_0$; which also holds true for mildly stable conditions. Without the stability functions $\psi_M$ and $\psi_H$ a cross sensitivity would still remain; however, not as intricate as

depicted in Fig. (4).

## 4   CONCLUSIONS AND DISCUSSION

The sensitivities of the meteorological pre-processor model MPP-FMI on its input values were examined by the means of algorithmic differentiation. The differentiation of the pre-processor was carried out by a source transformation AD tool called TAPENADE, yielding a program that

evaluates the desired sensitivity derivatives with machine precision accuracy. We focused on the evaluation of vertical fluxes in the atmosphere, and in particular on the sensitivity of the predicted inverse Obukhov length and friction velocity on the model input parameters. These two quantities were selected, as they are key parameters in view of air pollution.

The study shows that the predicted inverse Obukhov length and friction velocity are most sensitive

to wind speed, and second most importantly, to solar irradiation. The dependency on wind speed is most pronounced at low wind speeds. For both predicted inverse Obukhov length and friction velocity, the third most important factors are the roughness length and the surface albedo, for unstable and stable conditions, respectively. The surface roughness length determines, how sensitive the friction velocity is to wind speed.

The presented results have implications for improving the meteorological pre-processing models, and for selecting and preparing the measured input values for such models. For instance, the high sensitivity of the pre-processor to the values of the wind speed at the height of 10 m implies that the wind observations have to be selected very carefully. Clearly, the wind speed observations should be as representative as possible for the whole of the domain to be considered, and should not be



affected or substantially influenced by any local disturbances.

This study gave more confidence that AD in general, and the TAPENADE tool in particular are useful tools of assessment for studying quantitatively the ranges of sensitivities of the predicted parameters. The analysis is more comprehensive and versatile, compared with the use of previously applied sensitivity analysis methods. The sensitivities can be analysed for a wide range of initial

input conditions at minimal computation time expense.

The AD procedure is also useful for analysing the functioning of computer programs, and for improving their optimisation in terms of computing resources. In this study, all the dependencies of the predicted parameters on the model input values were found to be physically understandable and feasible. However, the procedure could also be useful for finding out potential inaccuracies of the

numerical solutions, or even mistakes in the structure of the computer codes.

The meteorological pre-processor parametrisation scheme (that is originally based on van Ulden and Holtslag) used in this study is in fairly common use in other countries within meteorological pre-processors and dispersion models. The initial conditions used in the model computations corresponded to the climate and environmental conditions in Helsinki. However, the range of

conditions at such a northern latitude vary substantially (for instance, the ambient temperatures were assumed to range from - 20 °C to + 30 °C), and the more moderate climatic conditions that are common for most of central Europe are actually included in the selected wide variability. The main insights and conclusions found out in this study are therefore probably similar for several other pre-processors used in Europe that use the same or a similar boundary layer scaling method.

Future research could address the determination of how the sensitivity of MPP-FMI impacts the modelled concentrations of pollutants. Such research could be done by source transforming a chain of models using AD, instead of only one model. The next chain of models to be investigated could be a combination of a meteorological pre-processor and an urban scale dispersion model. The sensitivity of the combined modelling system could also be evaluated in terms of other input values

of the dispersion model, in addition to the meteorological ones.

**CODE AVAILABILITY**

The source code for the meteorological pre-processor (MPP-FMI 3.0) is included in the supplementary material. The source-transformed code is also included in the supplementary

material. The source transformed code is subject to the TAPENADE licence agreement which limits the use of the code to academic research (see www-sop.inria.fr/tropics/tapenade/downloading.html).




The supplemental material also contains the code that was used to produce the input data and a wrapper to handle data input and output.

**APPENDIX A**

The empirical stability functions of Eq. (2) as implemented in the meteorological pre-processor are

$$\psi_M = (1 - 16\,z/L)^{1/4} - 1 \text{ for } L < 0$$
$$\psi_M = -17(1 - e^{-0.29z/L}) \text{ for } L > 0$$

(A1)

The stability functions of Eq. (A1) are taken from Karppinen et al. (1997). Figure A1 shows $u_*$ as a function of $L^{-1}$ for two different wind speeds (1 and 4 m s$^{-1}$). Note that $-L^{-1}$ and $L^{-1}$ are plotted on the same x-axis.


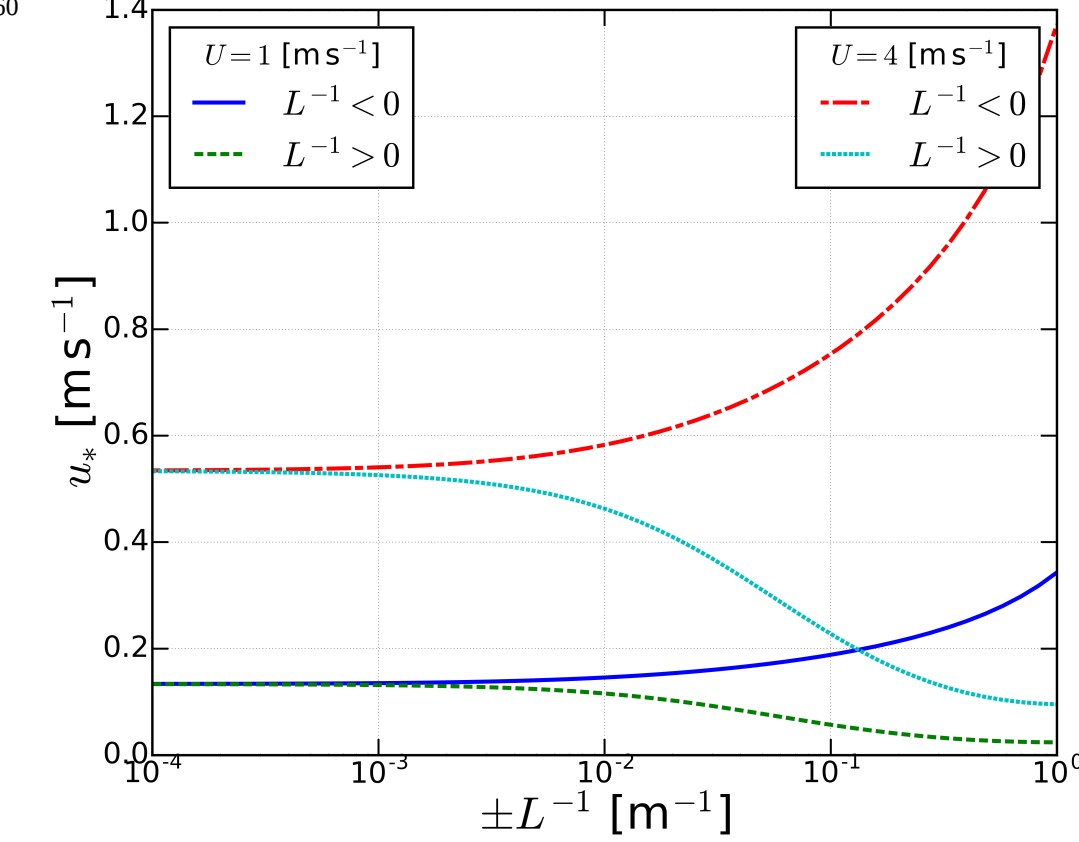

**Figure A1:** Friction velocity ($u_*$) as a function of inverse Obukhov length ($L^{-1}$) for two different wind speeds ($U$) using a roughness length ($z_0$) of 0.5 m and wind speed measurement height ($z$) of 10 m.



## APPENDIX B

This appendix covers the table lookup parameters that are used to estimate the surface albedo ($r$), Priestley-Taylor moisture parameter ($\alpha$).

The state of the ground is used in a table lookup to obtain an estimate for the surface albedo

according to surface type and the state of the ground. The table lookup procedure is shown in Table B1.

**Table B1:** Table lookup for surface albedo ($r$) based on surface type and state of the ground.

| | State of the ground | | | | | | | | | |
|---|---|---|---|---|---|---|---|---|---|---|
| | Soil | | | Ice | | Snow cover (%) | | | | |
| | Dry | Moist | Wet | Dry | Wet | <50 | 50<100 | 100 | 50<100 | 100 |
| **Surface** | | | | | | melting | melting | melting | dry snow | dry snow |
| Sea | 0.06 | 0.06 | 0.06 | 0.06 | 0.06 | 0.30 | 0.30 | 0.70 | 0.71 | 0.71 |
| Lake | 0.05 | 0.05 | 0.05 | 0.15 | 0.15 | 0.18 | 0.38 | 0.71 | 0.71 | 0.71 |
| Wasteland | 0.13 | 0.13 | 0.13 | 0.13 | 0.33 | 0.44 | 0.55 | 0.67 | 0.67 | 0.67 |
| Field | 0.2 | 0.2 | 0.2 | 0.13 | 0.11 | 0.33 | 0.55 | 0.67 | 0.67 | 0.67 |
| Forest | 0.11 | 0.11 | 0.11 | 0.11 | 0.17 | 0.26 | 0.34 | 0.39 | 0.39 | 0.39 |
| City | 0.22 | 0.22 | 0.22 | 0.13 | 0.11 | 0.17 | 0.22 | 0.28 | 0.28 | 0.39 |

The Priestley-Taylor parameter estimate is estimated using a table lookup involving weather codes,

solar elevation angle, state of the ground, and precipitation during the last 12 hours (Karppinen et al., 1997). The table lookup is illustrated by a flow chart depicted in Fig (B1).





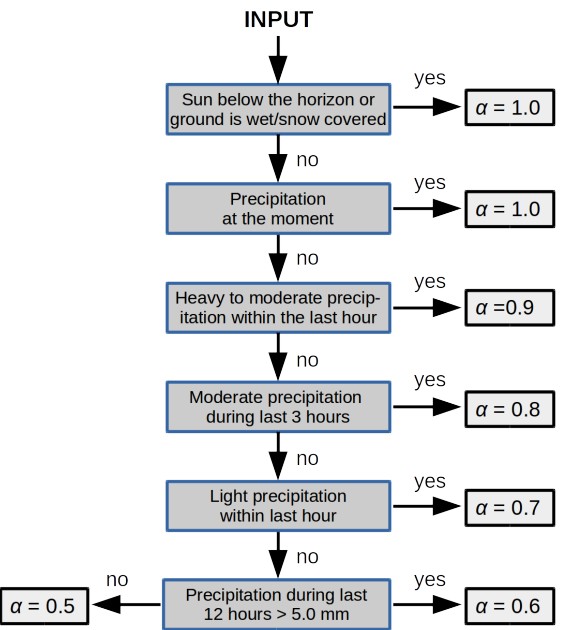

**Figure 1B:** Flow chart of how the Pristley-Taylor moisture parameter ($\alpha$) is estimated from input parameters that comprise state of the ground, current weather, weather during the last hour, weather during the last three hours, precipitation during last 12 hours, and solar elevation angle.

**Acknowledgements**

This work was funded by the Maj and Tor Nessling Foundation (grants 2014044 and 201600449).

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
