# Peer review of "Sensitivity analysis of the meteorological pre-processor MPP-FMI 3.0 using algorithmic differentiation"

_Geoscientific Model Development, 2016_

## Referee Comment (RC1) · L. Hascoet (Referee) · 24 Feb 2017

The submitted article deals with sensitivity analysis of a meteorological model, provided as a Fortran program. The derivatives used for this analysis are obtained by Source-Transformation Algorithmic Differentiation of the source of the program. The article provides an extensive discussion and physical interpretation of the obtained sensitivities. The discussion eventually leads to some recommendations on the measurements that are usually fed to the meteorological model. The article also provides a few suggestions on future extensions of the sensitivity study, for example by including more meteorological modules in the study.

Before giving my opinion, I'd like to point out that I'm not in a position to judge the

meteorological or physical aspects of the work. On the other hand, I can give my opinion on the computer science aspects and the use of Algorithmic Differentiation.

Overall, the paper is well organized and well written. The long part (section "Results") on the physical interpretation of the sensitivity results is probably obscure for outsiders but certainly meaningful for specialists. Still, I appreciate the visible effort that was devoted to present these results as clearly as possible. To my eyes, the interest of this paper is its illustration of the use of AD to sensitivity analysis in an Earth Sciences application.

From the viewpoint of AD tools, the paper could give more answers to a few questions, such as:

– Technical data: How does the runtime of the (tangent) differentiated code compare with the runtime of the original/primal code?

– The primal code being relatively short, did someone consider hand coding, and in that case how does the automatic AD code compare with hand-coded derivatives ?

– You mention that AD gives you machine accuracy (compared with divided differences), but the later discussion is based on figures 2,3,4 and probably doesn't need this accuracy all that much. Maybe the "accuracy" argument can be made stronger by pointing out that the choice of the "good" epsilon perturbation for divided differences is difficult and costly, especially when the orders of magnitude of the inputs are very different.

– I understand you selected tangent mode rather that reverse/adjoint mode, as you have 11 independents and 10 dependents. Your argument is slightly weakened by the fact that the results section concentrates only on two dependent outputs instead of 10. Nevertheless, your choice is still ok.

– Still, using the tangent mode, you need to run it 11 times at each data point, as you explain on page 7. I see from the provided files that you didn't use the "vector" tangent

mode, that could save 10 out of the 11 redundant executions of the primal instructions. Why is that ?

– Classically, when people want to compute full Jacobians (admittedly yours are a small enough 10*11) they try to exploit known sparsity of the Jacobian to compute it in a compressed way. Why didn't you do that? Maybe your Jacobian is not sparse? Then you might want to state that.

Other punctual remarks:

– Why was the radiosonde code not considered? Did it pose a problem to the AD tool ?

– You might reword slightly line 49: Tapenade is not the "only": OpenAD also pretty much fits.

– Line 51 is slightly misleading: readers might understand that AD produces the set of differentiated equations of the original math equations. We agree that if we consider the computer program as an alternative, roughly equivalent set of equations, then AD can be presented as producing the derivative equations of those alternative equations.

– On line 172: in fact the derivative instructions are always performed *before* the primal. The reason is quite anecdotical: think of the tangent diff of "y = x*y"

– Your statement on line 324 seems slightly optimistic: with or without AD, studying sensitivities at a large number of input data points is proportional to this number of points, and therefore not cheap. Not being a specialist, I suppose there might be ways to make it cheaper (surrogate models?) but they are clearly outside the scope of your study.

Typos:

Line 92: the the

Line 105: covarince

Line 149: a sequence of

---

## Referee Comment (RC2) · Anonymous Referee #2 · 28 Jun 2017

The article submitted by Backman et al. introduce an elegant approach to assess the sensitivity of a model in the context of small scale dispersion of air pollutants. It is proposed to use algorithmic differentiation (AD) to assess the sensitivity of the meteorological pre-processor of a dispersion model. The pre-processor evaluates crucial diagnostics to be used in the dispersion model such as the Obukhov lengths or the friction surface velocity. AD allows identifying and quantifying the sensitivity of those parameters to the variables provided in input. The scope, methodology and results are clearly explained. The source code is made available as well as its differentiated form, allowing a satisfactory reproducibility. Therefore, I support publication of this work in GMD provided that the following comments are considered.

[Figure]

Comments and questions

The article is limited to the sensitivity of the meteorological pre-processor, and deliberately avoids investigating the dispersion model itself. As such, the relevance of the results is somewhat limited, and the present article should be considered as a methodological proof of concept, which constitutes in itself an important building block, but leaves the reader expecting that the authors will pursue the efforts and include the dispersion model in the approach.

Extending the sensitivity analysis to dispersion modelling will undoubtedly raise the issue of the relative importance of drivers of mixing height in addition to Obukhov length and friction velocity. In the design of the meteorological preprocessor MPP-FMI, the mixing height is computed independently from the Obukhov length. It would be good to recall in Section 2.1 the rationale for this choice, and more specifically the consequences for the findings of the study. Mixing height is at least as important as Obukhov length and friction velocity in driving atmospheric dispersion in the surface layer and the matter should be discussed in more details. This comment regards both the methodological section, but also the results for instance in Section 3.3. on Cross Sensitivity, where the key findings should be put in perspective with the qualitative sensitivity that one might expected regarding mixing height (even if the quantitative sensitivity analysis is left outside of the scope of the paper).

L54: is it possible to assess the sensitivity to internal model parameters rather than input data using the AD approach?

L55: Further background information should be added regarding the fact that Tapenade proposes analytical derivatives for differentiable functions.

L148-150: There are computer programs that deal with non-derivable functions, how are those handled by AD? Isn't that the reason why in Section 3 (L192-194) the outcome of the outlook table is used instead of the (non-derivable) table itself ?

L157: please explain what is meant by "forward" or "reverse", and why the reverse mode should be favored in some cases (L182)

Technical comments

L40-44: provide the range of spatial scale for application of the mentioned models

L92: two occurrences of "the"

L149: a sequence "of" arithmetic

L185: provide the link for the web interface

L187-189: unclear sentence, rephrase

---

## Author Comment (AC1) · 30 Aug 2017

The authors would like to thank the reviewer for the constructive comments on how to improve the manuscript.

**General comments**

**Comment:** How does the runtime of the (tangent) differentiated code compare with the runtime of the original/primal code?

**Reply:** This is a good point and should definitely be included in the revised manuscript. The comparison was added to the revised manuscript as: "The source transformed

computer program was thus used to construct full Jacobian matrices and took just 4.5 times longer to run than the original program."

**Comment:** The primal code being relatively short, did someone consider hand coding, and in that case how does the automatic AD code compare with hand-coded derivatives?

**Reply:** It would indeed be feasible to hand code the derivative information into the original code and compare with the AD code. Although feasible, hand coding is, however, in the authors' opinion quite a tedious task for a code of this length. Since the present study is not focused on AD development or verification, hand-coding the derivative information was not pursued.

**Comment:** You mention that AD gives you machine accuracy (compared with divided differences), but the later discussion is based on figures 2,3,4 and probably doesn't need this accuracy all that much. Maybe the "accuracy" argument can be made stronger by pointing out that the choice of the "good" epsilon perturbation for divided differences is difficult and costly, especially when the orders of magnitude of the inputs are very different.

**Reply:** Again, a very good point. In the revised manuscript, these points are discussed as follows:
"The evaluation of finite differences is further complicated if input variables differs by orders of magnitude. By choosing the AD method, the tedious and imprecise evaluation can be avoided."
What is not visible from the figures, but discussed in writing, is that the stability parameter $L^{-1}$ can be very close to zero when the wind speed is high; hence, good accuracy is needed in those cases.

**Comment:** I understand you selected tangent mode rather that reverse/adjoint mode, as you have 11 independents and 10 dependents. Your argument is slightly weakened by the fact that the results section concentrates only on two dependent outputs instead

of 10. Nevertheless, your choice is still ok. Still, using the tangent mode, you need to run it 11 times at each data point, as you explain on page 7. I see from the provided files that you didn't use the "vector" tangent mode, that could save 10 out of the 11 redundant executions of the primal instructions. Why is that ?

**Reply:** This is a valid point. Since the code is not computationally that expensive the "vector" tangent mode was not initially used. In the revised manuscript, the differentiated code (and the wrapper) is done by exploiting the "vector" mode as suggested. The vector tangent mode is 2.4 times faster than the non-vector code in this case.

**Comment:** Classically, when people want to compute full Jacobians (admittedly yours are a small enough 10*11) they try to exploit known sparsity of the Jacobian to compute it in a compressed way. Why didn't you do that? Maybe your Jacobian is not sparse? Then you might want to state that.

**Reply:** The Jacobian is not sparse which is why the full Jacobian was constructed for each data point. This is now explicitly stated in the revised manuscript. Furthermore, it was not worth exploiting the sparsity that existed since the code is so quick to run anyway.

**Other punctual remarks**

**Comment:** Why was the radiosonde code not considered? Did it pose a problem to the AD tool?

**Reply:** It was not left out because of technical complications. The radiosonde data was left out because it does not affect the calculations of friction velocity nor the Obukhov length. The code that deals with radiosonde data is essentially a lookup procedure to find the temperature-inversion height from temperature and relative humidity data and is not interesting from a sensitivity point of view.

**Comment:** You might reword slightly line 49: Tapenade is not the "only": OpenAD also pretty much fits.

**Reply:** OpenAD is now also mentioned as an alternative.

**Comment:** Line 51 is slightly misleading: readers might understand that AD produces the set of differentiated equations of the original math equations. We agree that if we consider the computer program as an alternative, roughly equivalent set of equations, then AD can be presented as producing the derivative equations of those alternative equations.

**Reply:** The authors had missed the possibly misleading sentence which was also picked up by the other referee. To avoid confusion a more comprehensive explanation is now given which reads:
"A source transformation tool approaches the differentiation by analysing the source code of a given computer program and generating an augmented source code which contains, in addition to the original operations, instructions that carry out their chain rule differentiated versions. As these differentiated statements accompany each relevant mathematical operation in the source code, they propagate the derivative information across the entire program, providing exact sensitivity information (to machine precision) on how the inputs of the program influence its results."

**Comment:** On line 172: in fact the derivative instructions are always performed \*before\* the primal. The reason is quite anecdotical: think of the tangent diff of "y = x\*y"

**Reply:** This is of course true and was changed accordingly.

**Comment:** Your statement on line 324 seems slightly optimistic: with or without AD, studying sensitivities at a large number of input data points is proportional to this number of points, and therefore not cheap. Not being a specialist, I suppose there might be ways to make it cheaper (surrogate models?) but they are clearly outside the scope of your study.

**Reply:** Yes, the optimism needs to be downplayed. This relates to the earlier comment with the need to exploit sparse matrices to speed things up. The sentence was change

to

"The sensitivities could be analysed for a wide range of input conditions both accurately and effectively."

**Typos:**

**Comment:** Line 92: the the **Reply:** Corrected.

**Comment:** Line 105: covarince **Reply:** Corrected.

**Comment:** Line 149: a sequence of **Reply:** Corrected.

---

## Author Comment (AC2) · 30 Aug 2017

The authors would like to thank the reviewer for the constructive comments on how to improve the manuscript.

**Comments and questions**

**Comment:** The article is limited to the sensitivity of the meteorological pre-processor, and deliberately avoids investigating the dispersion model itself. As such, the relevance of the results is somewhat limited, and the present article should be considered as a methodological proof of concept, which constitutes in itself an important building block,

but leaves the reader expecting that the authors will pursue the efforts and include the dispersion model in the approach.

**Reply:** The choice to limit the study was indeed deliberate. The ultimate goal of, not only the presented meteorological pre-processor, but other meteorological pre-processors that are based on the Van Ulden and Holtslag (1985) publication, is indeed to provide parameters relevant for dispersion models. In the authors' opinion, the wide use of the method warrants restricting the study to meteorological pre-processing. The message that the manuscript focus on a meteorological pre-processor, and not in conjunction with a dispersion model, was clarified throughout the paper where needed, in light of the referee's comment.

**Comment:** Extending the sensitivity analysis to dispersion modelling will undoubtedly raise the issue of the relative importance of drivers of mixing height in addition to Obukhov length and friction velocity. In the design of the meteorological preprocessor MPP-FMI, the mixing height is computed independently from the Obukhov length. It would be good to recall in Section 2.1 the rationale for this choice, and more specifically the consequences for the findings of the study. Mixing height is at least as important as Obukhov length and friction velocity in driving atmospheric dispersion in the surface layer and the matter should be discussed in more details. This comment regards both the methodological section, but also the results for instance in Section 3.3. on Cross Sensitivity, where the key findings should be put in perspective with the qualitative sensitivity that one might expected regarding mixing height (even if the quantitative sensitivity analysis is left outside of the scope of the paper).

**Reply:** The mixing height is indeed computed separately from the Obukhov length, since the radiosonde routine uses the standard technique of potential temperature data from radiosondes to estimate the mixing height. The comparison of this profile method to methods where both friction velocity and Obukhov length are used in the mixing-height estimations is already available in literature (Karppinen et al. 2001). Indeed a future interesting study would certainly be on the relative importance of mixing

height, friction velocity, and Obukhov length to the dispersion of pollutants in a dispersion model, and the inter-relationship between them would not be so trivial, so we preferred to keep the manuscript concise without too much speculation. Nonetheless, in the revised manuscript mixing height is also highlighted as an important dispersion parameter. The end of the first paragraph of Section 2.1 now reads:

"However, we have not addressed the routines within the MPP-FMI model that deal with the vertical temperature gradient and hence mixing height which are obtained from temperature profiles provided by radiosondes (Karppinen et al. 2001). Mixing height is another key parameter for the modelling of dispersion of pollutants because it determines the spread of pollutants particularly vertically, and so any future dispersion-model sensitivity study, based on the present work, would naturally also use mixing height as an input."

It would surely be warranted to include mixing height to the discussion in Seciton 3.3 if mixing height was calculated in the fluxes routine (Fig. 1 in the manuscript). Since this is not the case, we refrained from adding speculation on the intricate nature of boundary layer evolution and stability to section 3.3, and instead kept text about mixing height more general.

**Comment:** L54: is it possible to assess the sensitivity to internal model parameters rather than input data using the AD approach?

**Reply:** Yes it is possible. One can rewrite the code so that the internal model parameters are inputs to the model. This will enable AD to add this model parameter to the Jacobian and thus enable the user to assess its impact on model output. This was in fact what is explained at the beginning of Results section when e.g. precipitation and state-of-the-ground inputs were replaced with the Priestley-Taylor moisture parameter.

**Comment:** L55: Further background information should be added regarding the fact that Tapenade proposes analytical derivatives for differentiable functions.

**Reply:** This was also raised by the other referee and the paragraph was rewritten to

provide more background information. The paragraph now reads:

"Other source transformation AD tools for Fortran are also available (e.g. OpenAD) and a representative list can be found from the community driven portal for algorithmic differentiation (http://www.autodiff.org). A source transformation tool approaches the differentiation by analysing the source code of a given computer program and generating an augmented source code which contains, in addition to the original operations, instructions that carry out their chain rule differentiated versions. As these differentiated statements accompany each relevant mathematical operation in the source code, they propagate the derivative information across the entire program, providing exact sensitivity information (to machine precision) on how the inputs of the program influence its results."

**Comment:** L148-150: There are computer programs that deal with non-derivable functions, how are those handled by AD? Isn't that the reason why in Section 3 (L192-194) the outcome of the outlook table is used instead of the (non-derivable) table itself?

**Reply:** The reason for omitting the table lookups was scientific and not technical. It is more informative to assess e.g. how the moisture parameter (that ranges from dry=0.5 to moist=1.0) affects the stability, than having the input as surface synoptic observations (SYNOP) codes (http://weather.unisys.com/wxp/Appendices/Formats/SYNOP.html). However, when using AD, keep in mind that partial derivatives of the output need not change when an input is changed if there is a table lookup (or rounding of real values) before a threshold is reached which results in a different value being returned from the table lookup (or rounding to a different value).

**Comment:** L157: please explain what is meant by "forward" or "reverse", and why the reverse mode should be favoured in some cases (L182)

**Reply:** The reverse mode will give one row of the Jacobian at a time. Thus, the reverse mode is much more effective if the number of inputs is much higher than the number

of outputs (rows). This is now also stated in the revised manuscript as:

"The reverse mode should be favoured when $n \gg m$ because the reverse mode constructs the Jacobian one row at a time and is therefore more efficient."

A more in-depth description of the difference between the two modes of AD does not seem motivated given the extent to which the description would have to be extended. Thus, the interested reader is referred to Griewank and Walther (2008) as cited in the manuscript.

**Technical comments**

**Comment:** L40-44: provide the range of spatial scale for application of the mentioned models

**Reply:** The spatial scale is urban, which is now mentioned in the revised manuscript.

**Comment:** L92: two occurrences of "the"

**Reply:** Corrected.

**Comment: L149:** a sequence "of" arithmetic

**Reply:** Corrected.

**Comment: L185:** provide the link for the web interface

**Reply:** The link is now provided.

**Comment:** L187-189: unclear sentence, rephrase

**Reply:** The sentence was rephrased to "In this work, if an input variable to the model was solely used in a table lookup, that input was replaced by the parameter that results from the table lookup".

**References**

Karppinen, A., Joffre, S. M., Kukkonen, J., and Bremer P.: Evaluation of inversion strengths and mixing heights during extremely stable atmospheric stratification, Int. J.

Environ. Pollut., 16, 1–6, doi:10.1504/IJEP.2001.000653, 2001.